https://doi.org/10.1038/s41467-020-15338-1　　**OPEN**

# Regulating strain in perovskite thin films through charge-transport layers

Ding-Jiang Xue [1,2,6], Yi Hou [1,6], Shun-Chang Liu[2], Mingyang Wei [1], Bin Chen[1], Ziru Huang [1], Zongbao Li[2,3], Bin Sun [1], Andrew H. Proppe [1,4], Yitong Dong[1], Makhsud I. Saidaminov[1], Shana O. Kelley [4,5], Jin-Song Hu [2] & Edward H. Sargent [1✉]

Thermally-induced tensile strain that remains in perovskite films following annealing results in increased ion migration and is a known factor in the instability of these materials. Previously-reported strain regulation methods for perovskite solar cells (PSCs) have utilized substrates with high thermal expansion coefficients that limits the processing temperature of perovskites and compromises power conversion efficiency. Here we compensate residual tensile strain by introducing an external compressive strain from the hole-transport layer. By using a hole-transport layer with high thermal expansion coefficient, we compensate the tensile strain in PSCs by elevating the processing temperature of hole-transport layer. We find that compressive strain increases the activation energy for ion migration, improving the stability of perovskite films. We achieve an efficiency of 16.4% for compressively-strained PSCs; and these retain 96% of their initial efficiencies after heating at 85 °C for 1000 hours—the most stable wide-bandgap perovskites (above 1.75 eV) reported so far.

[1] Department of Electrical and Computer Engineering, University of Toronto, Toronto, ON M5S 1A4, Canada. [2] Beijing National Laboratory for Molecular Sciences (BNLMS), CAS Key Laboratory of Molecular Nanostructure and Nanotechnology, Institute of Chemistry, Chinese Academy of Sciences, 100190 Beijing, China. [3] National Engineering Research Center for Advanced Polymer Processing Technology, Zhengzhou University, 450002 Zhengzhou, China. [4] Department of Chemistry, University of Toronto, Toronto, ON M5S 3G4, Canada. [5] Leslie Dan Faculty of Pharmacy, Department of Pharmaceutical Sciences, University of Toronto, Toronto, ON M5S 3M2, Canada. [6] These authors contributed equally: Ding-Jiang Xue, Yi Hou. ✉email: ted.sargent@utoronto.ca

The power conversion efficiencies (PCEs) of perovskite solar cells (PSCs) have increased from 3.8%[1] to a certified 25.2%[2], approaching those of crystalline silicon solar cells. Continued progress in stability remains a topic of importance along the path to industrial application[3–5]. Encapsulating PSCs to protect against oxygen and moisture is a straightforward method to stabilize these devices[6–8]; however, some important sources of instability are not overcome using extrinsic stabilization approaches such as encapsulation[9].

Residual tensile strain in perovskite films is a source of instability, and stems from the thermal expansion mismatch between perovskites and substrates during the annealing steps required in the formation of perovskites[10–13]. Specifically, thermally-induced tensile stresses in perovskite films can reach 50 MPa, sufficient to induce the deformation of copper[12]. These tensile stresses weaken bonds, decreasing the formation energy of defects and lowering the activation energy for ion migration[9,11]. These mechanisms are obstacles to stability at elevated temperatures, such as those required in accelerated lifetime testing.

The correlation between stress ($\sigma$) and thermal expansion mismatch is quantified as follows:

$$\sigma_{\Delta T} = \frac{E_p}{1 - v_p}(\alpha_s - \alpha_p)\Delta T \qquad (1)$$

where $E_p$ is the modulus of the perovskite, $v_p$ is Poisson's ratio in the perovskite, $\alpha_s$ and $\alpha_p$ are the thermal expansion coefficients of the substrate and the perovskite, respectively, and $\Delta T$ is the temperature gradient during cooling from the annealing temperature of the perovskite film to room temperature[12]. Several strategies have been reported to reduce this detrimental tensile strain in perovskite films based on the above equation[11,12]. They can be divided into two categories: (i) lowering the formation temperature of perovskite films to reduce $\Delta T$[12]; (ii) using plastic substrates with thermal expansion coefficients that are similar to that of the perovskite to decrease $\Delta \alpha$[11]. Although these strategies have been shown to diminish the residual tensile stress in perovskite films, they also lower device efficiency. Room-temperature-prepared $CH_3NH_3PbI_3$-based solar cells exhibit a PCE of 17.1%[14], a result of the lower quality of perovskite films fabricated by low-temperature processing. It is imperative to control the residual stress to move further in the direction of high efficiency and increased stability of PSCs.

Here we report a strain-compensation strategy that reduces the tensile strain in perovskite films with the aid of hole-transport layer (HTL). We use an HTL possessing numerous carbonyl anchoring groups that exhibit strong interactions with the perovskite surfaces, and thus we build a strong HTL: perovskite interface that transfers strain from the HTL to the perovskite active layer; we then balance the tensile/compressive strain transition by tuning the processing temperature and strain of the HTL. The resultant compressively-strained photovoltaic devices retain 95% of their initial PCEs of 16.4% after maximum power point tracking for 60 h, and 96% after heating at 85 °C for 1000 h, showing the most stable wide-bandgap perovskite (above 1.75 eV) cells reported so far. The improved stability of compressively-strained material stacks is explained through the higher activation energy (from 0.547 to 0.794 eV) for ion migration compared to that of tensile-strained films.

## Results
**Residual strain in perovskite films.** Based on Eq. (1), the difference in thermal expansion coefficients ($\alpha$) between the perovskite and the contacting layers provides one source of stress; and the requirement of high annealing temperatures to form crystalline perovskites also contributes due to a large $\Delta T$.

Regarding the thermal expansion mismatch, PSCs typically consist of a stack of multilayers including a substrate coated with a transparent conducting oxide electrode (TCO) layer, followed by the electron-transport layer (ETL), the perovskite layer, the hole-transport layer (HTL), and another electrode layer. The thermal expansion coefficient varies greatly with each functional layer (Fig. 1a). The widely used ITO-coated glass and metal oxide charge-transport layers possess low values of α in the range of 0.37 to $1 \times 10^{-5}$ K$^{-1}$[15]. In contrast, perovskites have much higher α values ranging from 3.3 to $8.4 \times 10^{-5}$ K$^{-1}$[16–18], approximately an order of magnitude higher than that of substrates or ETLs. Such a large thermal expansion difference is the main reason for the formation of tensile strain as perovskite films cool to room temperature[11,12].

As shown in Fig. 1b, the perovskite films used for high-efficiency perovskite solar cells typically require annealing at temperatures over 100 °C for improved crystallinity and minimization of defects[19–23]. Notably, when comparing with hybrid organic-inorganic perovskites, all-inorganic perovskites require even higher temperatures to stabilize the black cubic perovskite phase. In particular, $CsPbI_3$ requires annealing temperatures varying from 180 to 330 °C[19,24]. Therefore, inorganic perovskite films processed at high temperatures suffer from an even larger tensile strain.

When a perovskite film is deposited on a layer with lower α, the contact formed between the two layers during the high-temperature annealing process constrains the perovskite from contracting when it cools back to room temperature, introducing tensile strain along the in-plane direction (Fig. 1c). Conversely, when using a layer with higher α, the perovskite film contracts more, resulting in compressive strain. Therefore, tensile or compressive strain in perovskite films can be adjusted by using adjacent device layers with lower or higher α compared with that of perovskite.

We propose a strain-management approach to reduce residual tensile strain in perovskite films—since tensile strain is more often produced in these layer stacks than compressive strain. When using a substrate with lower α, we are able to compensate the tensile strain through an external compressive strain that is induced by the top contacting HTL with higher α, leading to a non-strained or even compressively-strained perovskite film.

**Strain-compensated perovskite films.** To evaluate this strain-compensation strategy experimentally, we focused on all-inorganic cesium lead halide perovskites primarily due to their superior thermal stability compared to their organic-inorganic hybrid counterparts[19,20]. This high thermal stability enables all-inorganic perovskites to withstand higher annealing temperatures (Fig. 1b), while also inducing a large amount of stress considering the soft lattice of perovskites that can endure large strain (Fig. 1d and Supplementary Fig. 1), thereby facilitating the investigation of strain control. Specifically, we chose $CsPbI_2Br$ as a suitable candidate to study the strain engineering in perovskite films since it is a more stable photoactive perovskite cubic structure, and it provides a wide bandgap of 1.92 eV[25–28], potentially serving as top cells in tandem devices with silicon or narrow-bandgap perovskite solar cells.

We took the view that functional layers suitable for our strain-compensation strategy would need to meet the following three criteria: (i) the top functional layer should have a higher thermal expansion coefficient compared with the perovskite, offering the possibility of compressive strain; (ii) the functional layer should have a strong interaction with the perovskite in order to anchor to the lattice and achieve strain offset; (iii) the top interface layer should be coated at high temperatures, inducing a compressive

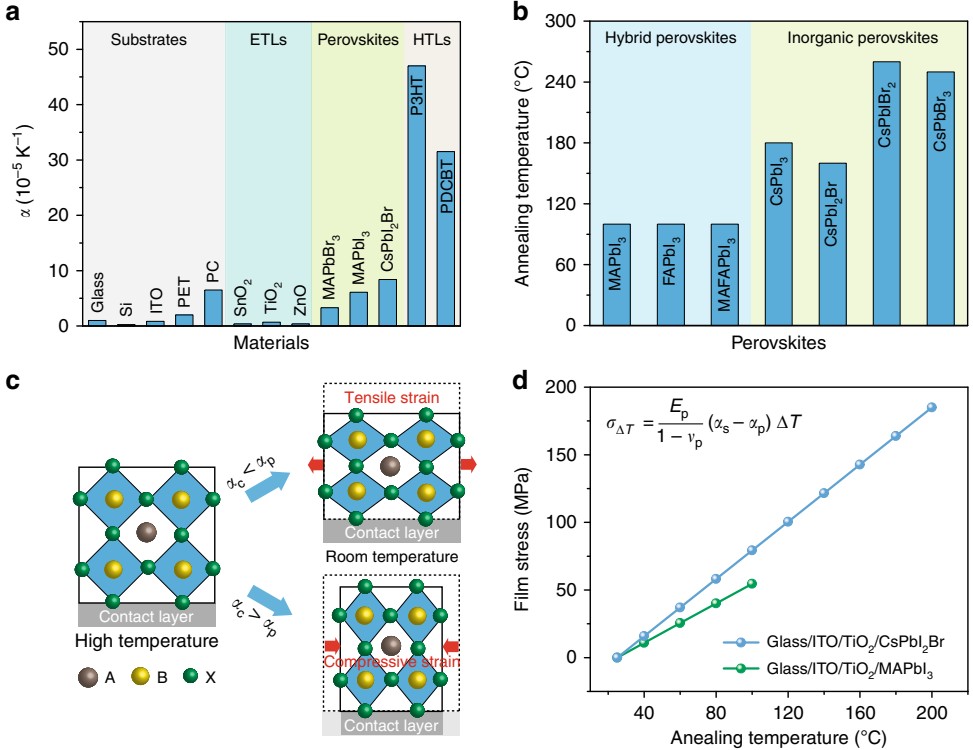

**Fig. 1 Analysis of thermally-induced strain in perovskite films. a** Thermal expansion coefficients of widely-used functional layers in PSCs including substrates, ETLs, perovskites, and HTLs. **b** Annealing temperatures of different hybrid and inorganic perovskite films during formation. **c** Schematic showing the formation of tensile and compressive strains. **d** The calculated annealing temperature-dependent stress from thermal expansion mismatch between the substrate and perovskites including MAPbI$_3$ and CsPbI$_2$Br.

strain when cooling back to room temperature, according to the above equation relating stress and thermal expansion mismatch.

In this regard, we believed that poly[5,5-bis(2-butyloctyl)-(2,2-bithiophene)-4,4′-dicarboxylate-alt-5,5′-2,2′-bithiophene] (PDCBT) with rich carbonyl groups, a well-known polymer that can serve as a highly efficient HTL for PSCs[29,30], and with a similar chemical structure to poly(3-hexylthiophene-2,5-diyl) (P3HT) (Supplementary Fig. 2), fulfilled the above criteria and would offer a compressive strain to offset the residual tensile strain in perovskite films. We first performed temperature-dependent X-ray diffraction (XRD) to measure the thermal expansion coefficient of PDCBT. The $\alpha$ of PDCBT was calculated to be $31.5 \times 10^{-5}\,\mathrm{K}^{-1}$ through the shift of XRD peaks (Fig. 2a and Supplementary Fig. 3). This value was close to that of P3HT[31–33] and more than an order of magnitude higher than that of CsPbI$_2$Br (Fig. 2a and Supplementary Fig. 4), providing a large thermal expansion mismatch with the perovskite.

We then carried out X-ray photoelectron spectroscopy (XPS) to evaluate the interaction between PDCBT and the perovskite. Due to the shallow probing depth of XPS (about 10 nm), a thin layer of PDCBT (less than 8 nm) was coated on the top of perovskite film for the measurement. We observed that the Pb 4$f$ peaks shifted towards lower binding energy after PDCBT coating (Fig. 2b). In addition to the O 1$s$ peak from PDCBT at 531.5 eV (as observed in pure PDCBT films), there is another O 1$s$ peak from perovskite/PDCBT film, which shifted towards a higher binding energy at 532.9 eV (Fig. 2c). The S 2$p$ peaks remained constant when compared with pure PDCBT (Fig. 2d). Therefore, the XPS results demonstrated the strong interaction between the perovskite and PDCBT through the formation of Pb–O bonds, consistent with our electrostatic potential analysis calculated using density functional theory (DFT), where the electron-withdrawing atom was O, providing favorable condition for the

passivation of under-coordinated Pb in the perovskite[34,35] (Supplementary Fig. 5). Moreover, the introduction of PDCBT resulted in an evident blue-shifted photoluminescence (PL) emission peak for the perovskite film compared to that of P3HT and Spiro-OMeTAD (Fig. 2e), further indicating the efficient passivation effect of PDCBT while confirming the strong binding between perovskite and PDCBT[36], in good agreement with XPS results.

We carried out further experiments varying the coating temperature of PDCBT layer, a key processing parameter for our strain-compensation strategy. Cl-capped TiO$_2$ (TiO$_2$-Cl) nanocrystal films were used as an ETL according to previous work[37,38]; CsPbI$_2$Br films were formed by annealing at 160 °C for 10 min (Supplementary Fig. 6), resulting in a tensile stress of about 150 MPa (Supplementary Fig. 1). The possible strain induced by local lattice mismatches can be excluded from the observed compositional homogeneity in CsPbI$_2$Br films along the lateral (Supplementary Fig. 7) and vertical directions (Supplementary Fig. 8), as well as the little variation of lattice constant from the surface to the bottom (Supplementary Fig. 9). The residual strain in the perovskite films was insensitive to post-annealing treatment due to the strong adhesion between perovskite and substrate once the perovskite is formed, according to a previous report[11]. Therefore, according to our strain-compensation strategy, when coating PDCBT at high temperatures, the resulting compressive strain can offset the residual tensile strain. Figure 2f shows a linear correlation between stress and coating temperature of the PDCBT. This is consistent with a reduction of tensile stress in the perovskite film when increasing the HTL coating temperature.

To verify this hypothesis, we used out-of-plane XRD to probe the total strain in perovskite films after depositing the HTL at different temperatures, evaluated through the XRD peak shift.

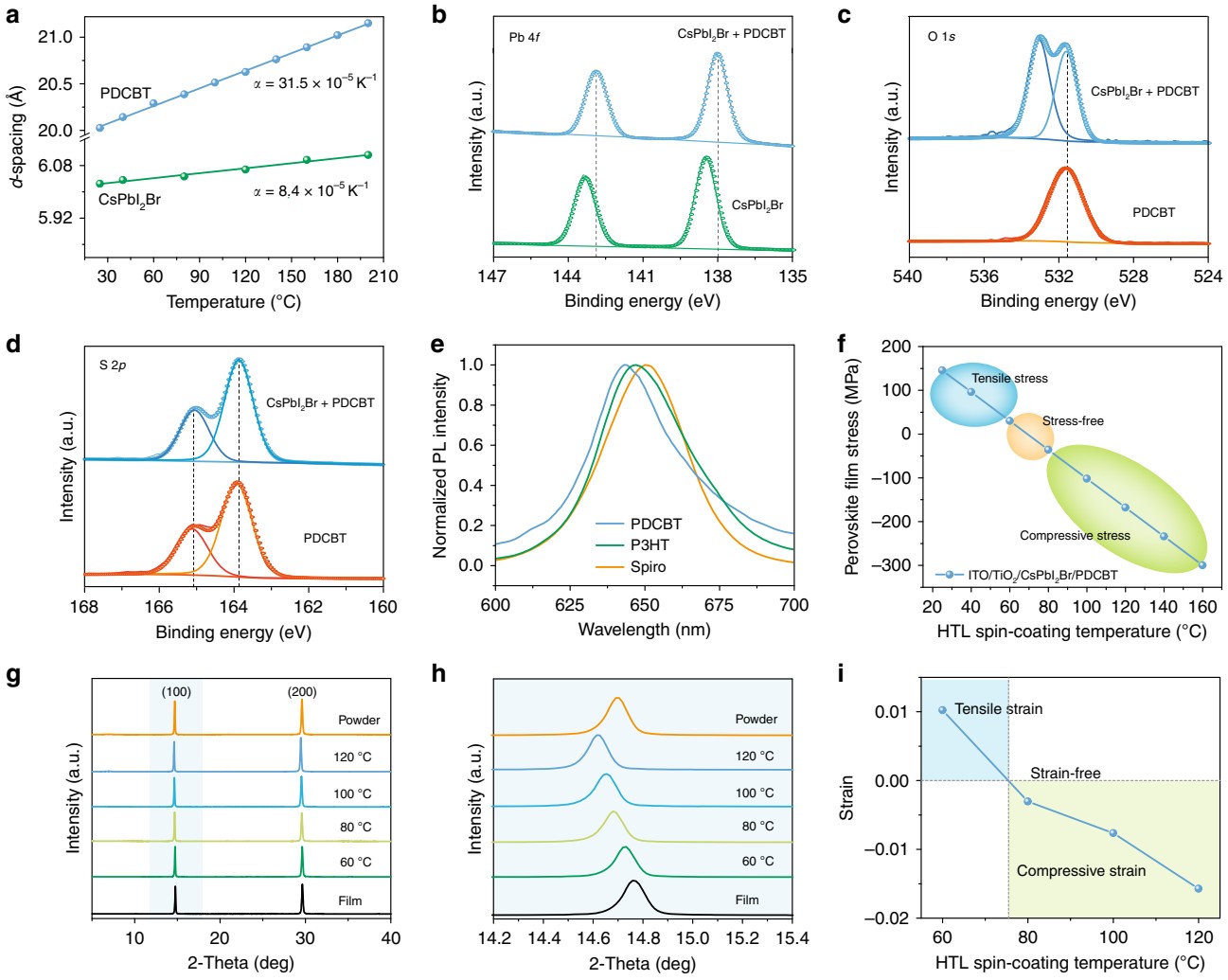

**Fig. 2 Characterization of strain compensated perovskite thin films. a** Temperature-dependent (100) d-spacing of PDCBT and perovskite films. XPS spectra of (**b**) Pb 4f, (**c**) O 1s and, (**d**) S 2p in the perovskite, perovskite/PDCBT, and PDCBT films. **e** Normalized PL spectra of perovskite films with PDCBT, P3HT and Spiro-OMeTAD from the ITO side. **f** The calculated net average stress in perovskites within structures consisting of ITO/TiO₂/perovskite/PDCBT as a function of PDCBT spin-coating temperature. **g** XRD patterns of perovskite film, powder and perovskite/PDCBT films fabricated at different PDCBT spin-coating temperatures. **h** Magnified (100) diffraction peaks in the region indicated by the blue. **i** Measured strain in perovskite films coated with PDCBT HTLs at different spin-coating temperatures.

The XRD peaks of as-fabricated perovskite film shifted to higher diffraction angles compared to the strain-free peaks of scraped perovskite powder (Fig. 2g). Such a shift to higher angles can be attributed to the smaller plane spacing in the direction perpendicular to the substrate, thereby indicating the tensile strain in the horizontal direction of the film based on the positive Poisson's ratio in perovskites, in good agreement with previous reports[9,11,12]. Moreover, a gradual shift to lower diffraction angle is observed, as expected, and is shown in the magnified view of (100) peaks when increasing the HTL coating temperature (Fig. 2h and Supplementary Fig. 10). Note that the strain was nearly eliminated when the HTL coating temperature was approximately 80 °C. As the temperature is increased to 120 °C, the film becomes compressively strained (Fig. 2i). Furthermore, the comparison of XRD peaks between the perovskite/PDCBT film and the same perovskite film after the PDCBT is washed away indicates that the compressive strain stems from the PDCBT layer coated at high temperature (Supplementary Fig. 11). The depth-dependent grazing incidence XRD (GIXRD) results indicate the homogeneous distribution of compressive strain in this perovskite film (Supplementary Fig. 12). Consequently, these

results demonstrate that the residual tensile strain in the perovskite film can be compensated by depositing the HTL atop the perovskite at high temperatures to introduce compressive strain, resulting in a strain-free or even compressively-strained perovskite film.

**Stability of perovskite films under different strains**. Such tunable strain in perovskite films, by controlling tension and compression, allows us to better investigate the correlation between film stress and the resulting stability. Despite the fact that perovskite vacancies are shallow electronic traps that do not act as recombination centers, they have been recently shown to strongly contribute to perovskite decomposition[37]. Vacancy-assisted migration of halide ions results in halide segregation in mixed-halide perovskites, and subsequently phase segregation[9].

We first calculated the formation energies of halide vacancy under different strain conditions using DFT, which directly reflected the density of halide ion vacancies in perovskite. We found that the tensile strain decreased the formation energy of halide vacancies, while compressive strain increased their

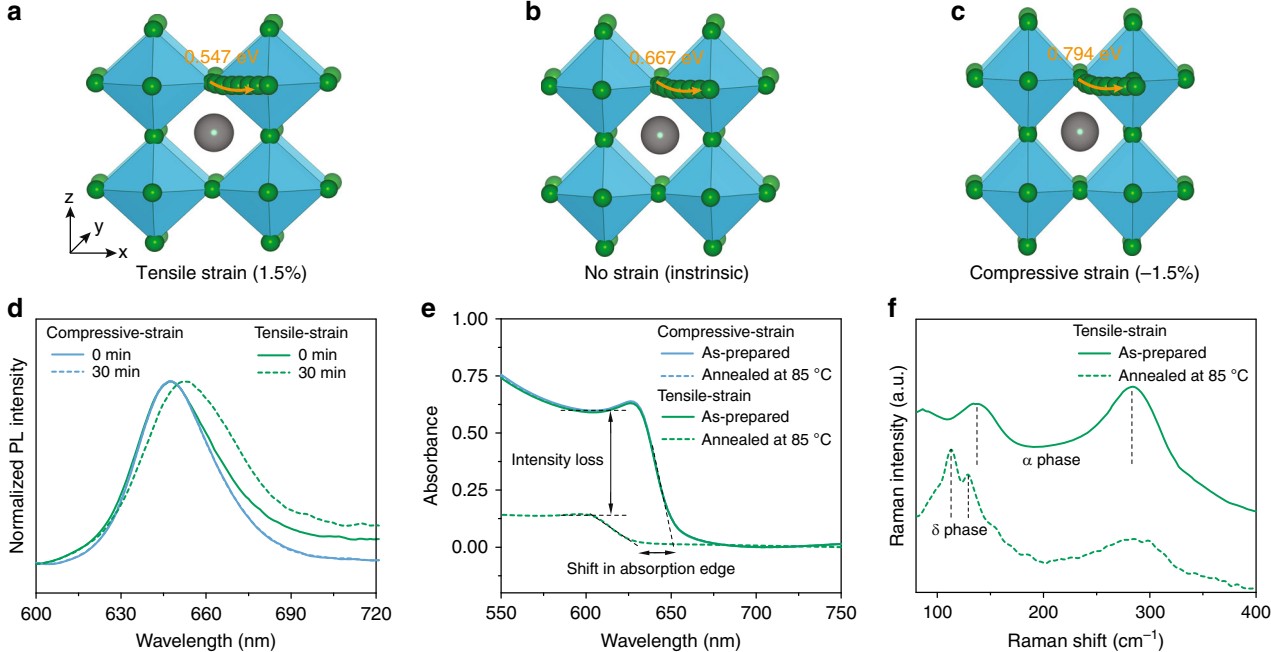

**Fig. 3 Stability of perovskite films under different strains.** Calculated activation energies for halide ion migration in perovskites under biaxial (**a**) 1.5% tensile strain, (**b**) no strain and (**c**) −1.5% compressive strain. **d** Normalized PL spectra of compressive-strain and tensile-strain perovskite films after illuminating for 0 and 30 min. **e** Absorption spectra of compressive-strain and tensile-strain perovskite films before and after annealing at 85 °C for 60 h. **f** Raman spectra of tensile-strain perovskite film before and after annealing at 85 °C for 60 h.

formation energy compared with intrinsic (strain-free) perovskites (Supplementary Fig. 13). Once halide vacancies are formed, we further calculated the relative activation energies for the vacancy-assisted migration of halide ions ($E_a$) in perovskites under biaxial tensile and compressive strains (Supplementary Fig. 13 and 14). The activation energy for halide ion migration in strain-free perovskite was 0.667 eV (Fig. 3b), agreeing well with previous reports[39,40]; a lower activation energy of 0.547 eV was found under biaxial tensile strain of 1.5% (Fig. 3a), whereas a higher energy of 0.794 eV was found for biaxial compressive strain of −1.5% (Fig. 3c). The above results indicate that compressive strain not only suppresses the formation of halide vacancies, but also decelerates ion migration in perovskites, improving their intrinsic stability.

To investigate the effect of strain on the stability of perovskite films from experiment, we studied the impact of strain in perovskite films on their rate of degradation. Mixed-halide $CsPbI_2Br$ perovskite films are known to undergo phase segregation into separated Br-rich and I-rich phases when exposed to light and heat, a consequence of increased ion migration activated under these conditions (Supplementary Fig. 15). This facilitated the measurement of halide ion migration[21], which can be readily investigated through optical characterizations of Br-rich and I-rich phases.

We first investigated the photostability of perovskite films under compressive and tensile strains. According to Fig. 2e, the perovskite films coated with PDCBT at 120 °C and 60 °C were labeled as compressive-strain and tensile-strain perovskite films, respectively. After 30 min of continuous illumination, the compressive-strain film showed no change of PL peak position, whereas the tensile-strain film exhibited a redshift in the peak position (Fig. 3d), a signature of phase segregation. This improved photostability indicates that the compressive-strain film exhibits suppressed halide ion migration, in good agreement with our calculated results.

We then explored the thermal stability of compressive-strain and tensile-strain perovskite films. For the tensile-strain films,

there are clear changes in the absorbance amplitude and a shift of absorption edge after annealing at 85 °C in air for 60 h (Fig. 3e). This blueshift can be attributed to formation of Br-rich phase, while the loss absorption amplitude was due to the phase transition of I-rich phase from black α phase to transparent δ phase, indicative of severe halide ion migration at high temperatures[21]. Raman spectroscopy further confirmed the coexistence of α phase and δ phase perovskites[22] (Fig. 3f). For compressive-strain film, the absorption spectrum remained unchanged after the same annealing process, exhibiting diminished ion migration under compressive strain. These findings agreed with our DFT results that compressive strain can suppress ion migration owing to high activation energy, improving the intrinsic stability of perovskites, similar to previous reports[9,11,12].

**Stability improvement in compressive-strain PSCs.** To further investigate the effect of strain on device performance, three types of PSCs under different strains were fabricated with processing temperatures of the PDCBT HTL varied from 60 °C (tensile strain), 90 °C (strain free), and 120 °C (compressive strain). We used a planar architecture of $ITO/TiO_2-Cl/perovskite/PDCBT/MoO_x/Au$ (Fig. 4a). Figure 4b showed the statistical photovoltaic performance of the PSCs prepared in these three different conditions.

Compared with the tensile-strain and strain-free devices with average PCEs of 14.8 and 15.5%, the compressive-strain devices exhibited higher average PCEs of 16.0%. The best-performing devices of each type demonstrated AM1.5 PCEs of 15.14, 16.00, and 16.41% (Fig. 4c and Supplementary Fig. 16). Considering the wide bandgap of $CsPbI_2Br$ (1.92 eV), the performance of our compressive-strain devices is comparatively superior to that of strain-free $CH_3NH_3PbI_3$-based devices with a bandgap of 1.64 eV fabricated at room temperature[14]. Notably, all the devices exhibited high FFs above 80%, especially the compressive-strain device with 85.1%, comparable to those of the best-reported

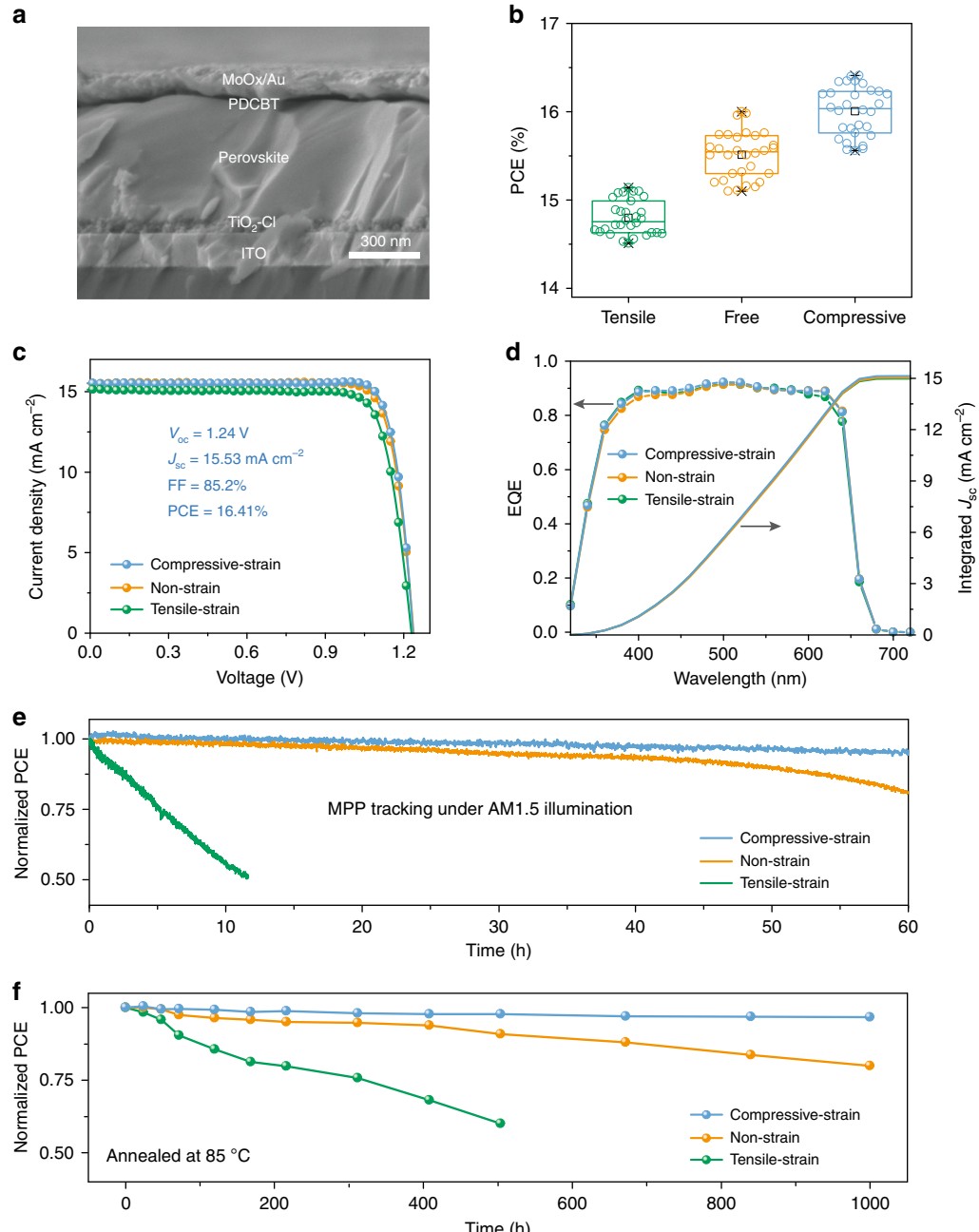

**Fig. 4 Performance of PSCs under different strains. a** Cross-sectional SEM image of PSCs. The scale bar is 300 nm. **b** PCE statistics of 30 PSCs under each strain condition. The boxes indicate the 25th and 75th percentiles. The whiskers indicate the 5th and 95th percentiles. The cross symbols represent the maximum and minimum values. The median and mean are represented by the line dividing the boxes and the open square symbols, respectively. **c** J-V curves of compressive-strain, strain-free and tensile-strain PSCs. **d** EQE curves of compressive-strain, strain-free and tensile-strain PSCs. **e** Evolution of normalized PCEs under MPP tracking and continuous simulated solar illumination (100 mW cm$^{-2}$). **f** Evolution of normalized PCEs of PSCs kept at 85 °C in a nitrogen atmosphere.

PSCs[41–43], further confirming the efficient passivation and carrier extraction efficiency of PDCBT. The integrated photocurrent densities from EQE spectra (Fig. 4d) were consistent with the $J_{sc}$ values measured from J-V characterization (within 5% deviation).

Despite the comparable performances of PSCs under different strains, we found that the photostability under maximum power point (MPP) conditions and continuous one sun illumination varied greatly with strain (Fig. 4e). The compressive-strain and non-strain devices maintained 95 and 81% of their initial PCEs after continuous MPP operation for 60 h, whereas the tensile-strain device showed a rapid loss of its 50% initial PCE only after

tens of hours, consistent with the observed photostability of perovskite films under different strains. We further tested the thermal stability of PSCs under continuous heating at 85 °C. The stability evolution was similar to the case under MPP condition (Fig. 4f). The tensile-strain device experienced significant PCE loss, while the compressive-strain and non-strain devices retained 96 and 80% of their initial PCEs after 1000 h of heating at 85 °C. The PCE loss of the tensile-strain device may be attributed to the phase segregation into Br-rich and I-rich perovskite phases, and subsequent transition of I-rich phase from photoactive to non-photoactive phase (α to δ)[21]. This result agrees with the observed

thermal-stability of perovskite films under different strains[11,12]. Therefore, the extended photostability and thermal-stability of compressive-strain PSCs supports the contention that compressive strain can increase the activation energy for ion migration and suppress the halide ion migration in mixed-halide perovskites.

## Discussion

In summary, we have introduced a strain-compensation strategy to reduce the thermally-induced tensile strain in perovskite films, by applying an external compressive strain produced by top functional layer. The high thermal expansion coefficient, strong binding affinity for the perovskite layer, and high processing temperature of the polymer HTL were found to be crucial parameters for this strain compensation strategy. Perovskite films with tunable strain (tensile to compressive) were successfully fabricated by adjusting the processing temperature of the HTL. We found that compressive strain can improve the intrinsic stability of perovskites, stemming from an increased activation energy for ion migration, and further enhance the photostability and thermal stability of state-of-art PSCs. This study provides an approach to regulate the strain in perovskite films, towards efficient and stable perovskite optoelectronic devices.

## Methods

**Solar cell fabrication**. Pre-patterned indium oxide (ITO, TFD Devices)-coated glass slides were ultrasonically cleaned using acetone and isopropanol for 45 min each. The Cl-capped $TiO_2$ ($TiO_2$-Cl) nanocrystal solution was synthesized according to previous work[38]. The $TiO_2$-Cl layer was spin-coated twice on ITO substrate at 3000 rpm for 30 s and then annealed on a hot plate at 150 °C for 30 min in ambient air. The substrates were immediately transferred to the $N_2$-filled glovebox after cooling to room temperature. To make the 1.5 M $CsPbI_2Br$ precursor solution, CsI, $PbI_2$ and $PbBr_2$ were mixed with a mole ratio of 2:1:1 in DMSO. The perovskite films were deposited onto the $TiO_2$-Cl substrates with a two-step spin-coating procedure. The first step was 1000 rpm for 10 s with an acceleration of 200 rpm s$^{-1}$. The second step was 3000 rpm for 90 s with a ramp-up of 1500 rpm s$^{-1}$. The coated films were then annealed on hotplates in two steps: (i) 45 °C for 3 min; (ii) 160 °C for 10 min. PDCBT was dissolved in a mixture of chlorobenzene (CB) and 1,2-dichlorobenzene (ODCB) (1:1 by volume) at a concentration of 10 mg/mL, and followed by stirring on a 60 °C hot plate for 30 min. For the PDCBT hot-casting process, the prepared perovskite film was first preheated to a desired temperature (60–120 °C) on a hot plate for 10 min, and the PDCBT solution was also kept on a hot plate for 10 min at the same temperature as the perovskite film. The hot perovskite film was then immediately (within 5 s) transferred to the spin coater chuck (which was at room temperature), and the hot PDCBT solution was dropped onto the hot film using a pipette almost simultaneously (Supplementary Fig. 17). The spin-coater was immediately started at a spin speed of 1000 rpm for 20 s. The thickness of PDCBT layer was about 50 nm (Supplementary Fig. 18). Finally, a 20 nm layer of $MoO_x$ and 100 nm layer of Au counter electrode were deposited on top of PDCBT by thermal evaporation in an Angstrom Engineering deposition system.

**Materials and device characterization**. XRD patterns were collected using a Rigaku D/Max-2500 diffractometer equipped with a Cu Kα1 radiation (λ = 1.54056 Å). XPS measurements were performed on an ESCALab220i-XL electron spectrometer (VG Scientific) using 300 W Al Kα radiation. Raman spectra (Horiba JobinYvon, LabRAM HR800) were measured under the excitation line of 532 nm. Optical absorption measurements were carried out in a Lambda 950 UV/Vis spectrophotometer. PL spectra were measured using a Horiba Fluorolog time correlated single-photon-counting system with photomultiplier tube detectors. The excitation source was a laser diode at a wavelength of 374 nm. The pulse duration is 110 ps, and peak power per area is 540 W cm$^{-2}$. The light was illuminated from the glass side of the perovskite film. SEM images and EDS mapping were collected using Hitachi SU5000. For EDS mapping, the acceleration voltage was set at 5 kV with spot intensity of 30 to increase the EDS signal to noise ratio. For secondary electron imaging, the acceleration voltage was kept at 3 kV with spot intensity of 10 to minimize the beam damage. The J-V characteristics of all the devices were measured using a Keithley 2400 source meter under AM1.5 G illumination (Newport, Class A). Unless otherwise stated, the J-V curves were measured in a $N_2$ atmosphere with a scanning rate of 50 mV s$^{-1}$ (voltage step of 10 mV and delay time of 200 ms). The active area was determined by the aperture shade mask (0.049 cm$^2$) placed in front of the glass side of the solar cell to avoid overestimation of the photocurrent density. EQE measurements were performed using a Newport system (QuantX 300) with monochromatic light without any bias light. The system was

calibrated by a certified Si solar cell (Newport) before the EQE measurement. The stability test was carried out at continuous MPP operation under AM1.5 G illumination by fixing the voltage at VMPP and then tracking the current output with a 420 nm cutoff UV filter. Thermal long-term stability tests were carried out by repeating the J-V characterizations over various times. The devices were kept at 85 °C in a nitrogen-filled glovebox.

**Simulation methods**. We used density-functional theory (DFT) performed in Vienna Ab Initio Simulation Package (VASP[44]) to study the perovskite crystal structure and migration of halide ions in $CsPbI_2Br$. Generalized gradient approximation (GGA) was considered with Perdew-Burke-Ernzerhof (PBE) generalized exchange-correlation functional[45]. The vdW correction of Grimmer's DFT + D2[46] was used to describe the weak interactions within the perovskite material while the plane-wave kinetic energy cutoff was fixed at 400 eV. The computational cell consisting of 64 $CsPbI_2Br$ units (320 atoms) was employed in all calculations. For structure relaxation, forces were converged to less than 0.01 eV Å$^{-1}$. The Brillouin zone was sampled using a gamma-only wavevector grid and an electronic convergence criterion of $5 \times 10^{-7}$ eV per formula unit was used. The halide vacancy defects were created by removing one halide atom from the 64 formula-unit computational cells. Activation energies for diffusion processes under different biaxial strains were computed from the total energy difference between diffusing species in the ground-state configuration and at the saddle point of the diffusion process[40]. The migration mediated by halide ion vacancies was examined using nudged elastic band and constrained energy minimization methods. The migrating species was propagated along the migration direction in a series of small steps with all unconstrained degrees of freedom relaxed at each step.

**Reporting summary**. Further information on research design is available in the Nature Research Reporting Summary linked to this article.

## Data availability

The data that support the findings of this study are available on reasonable request from the corresponding author.

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

## Acknowledgements
This publication is based in part on work supported by the US Department of the Navy, Office of Naval Research (Grant Award No. N00014-17-1-2524), the Ontario Research Fund Research Excellence Program, and by the Natural Sciences and Engineering Research Council (NSERC) of Canada. D.-J.X. acknowledges the support of the National Natural Science Foundation of China (21922512, 21875264), and the Youth Innovation Promotion Association CAS (2017050).

## Author contributions
D.-J.X. and Y.H. conceived the idea, prepared films, fabricated all devices and characterized them. S.-C.L. assisted in X.R.D. and X.P.S. measurements. M.W., B.C., and Z.H. assisted in PL and SEM measurements. Z.L. performed the DFT calculations and analyzed the results. B.S., Y.D., and M.I.S. assisted with the discussions. S.O.K. and J.-S.H. assisted with the manuscript preparation. E.H.S. directed the overall research. D.-J.X., Y.H., A.H.P., and E.H.S. wrote the manuscript. All authors read and commented on the manuscript.

## Competing interests
The authors declare no competing interests.
