## [Peer Review File · Nature Communications]

REVIEWERS' COMMENTS:

Reviewer #1 (Remarks to the Author):

This manuscript describes a strategy to compensate residual tensile strain in perovskite films by introducing an external compressive strain from the hole-transport layer (HTL). The perovskite solar cells (PSCs) achieved a power conversion efficiency (PCE) of 16.4% and retain 96% of their initial PCEs after heating at 85 °C for 1000 hours – the most stable wide-bandgap perovskites (> 1.75 eV) reported so far. Strain engineering is important to further promote the performance of perovskite solar cell. The result is exciting and the strategy to compensate residual tensile strain is novel.

Therefore, I suggest accepting the manuscript after some concerns are addressed as follows.

1. The authors described that they used a substrate with lower coefficient of thermal expansion (α) and the HTL with higher α . So, after annealing, because of the α mismatch between functional layer and substrate, the HTL would bear a tensile strain due to substrate constraint, at the same time the perovskite film would also bears tensile strain. Since both of perovskite film and HTL bear tension from substrate, how can the HTL compensate residual tensile strain in perovskite films? It requires further explanation.

Considering the PDCBT hot-casting process. the prepared perovskite film was first preheated to a desired temperature (60 C-120 C) on a hot plate. Is it the heating process that affects the strain in perovskite films?

2. In the manuscript the correlation between stress (σ) and thermal expansion mismatch is quantified as formula (1):

$$\sigma_{\Delta T} = \frac{E_p}{1 - \nu_p} (\alpha_s - \alpha_p) \Delta T$$

This formula can be used to calculate the stress due to thermal expansion mismatch between monolayer and substrate, is it be appropriate for calculating the stress due to thermal expansion mismatch between bi-layers (perovskite and HTL) and substrate? I think maybe it is better to adopt equations for accurate multilayer estimation.

3. What is the thickness of PDCBT layer? In figure 4(a), I can evaluate the thickness of PDCBT layer is thinner than TiO₂-Cl layer, about several tens of nanometers. The authors attribute XRD peaks shift to the compressive strain from the HTL. It is doubted that the HTL layer with this thickness can affect the whole perovskite layer that is much thicker and stiffer.

I suggest the authors give more evidence to support that the XRD peaks shift is mainly comes from the compressive strain from the HTL. If the XRD peaks shift was mainly comes from the compressive strain from the HTL, can the XRD peaks of perovskite film reverse to tension strain state after the HTL layer is washed out? If possible, the grazing incidence XRD measurement can be used to explore the surface strain change of perovskite film after HTL layer deposition.

4. In the DFT simulation, it lacks details with respect to the strain states. Is it under triaxial tension or biaxial? It is better to stimulate the strain state that exhibit in the perovskite film experimentally obtained.
5. In supplementary Figure 1e, the strain under 240 °C is about 0.05, such a large strain can fracture the steel, is it reasonable? Please check the number, or it requires a careful explanation.
6. I suggest that the J-V curves under forward and reverse bias of compressive-strain, strain-free and tensile-strain PSCs should be added in supplementary information for the figure 4c.
7. 3 On Page 14, line 331, what is the specific experiment condition of the thermal stability of PSCs under continuous heating at 85 °C, for example, the humidity of storage environment.

Reviewer #2 (Remarks to the Author):

In this manuscript, the authors demonstrate a strain-compensation strategy to reduce the tensile strain in perovskite films with the aid of top layer (PDCBT). The authors claim this method can balance the tensile/compressive strain transition by tuning the processing temperature and strain of the top layer (PDCBT). The PSCs have a champion PCE of 16.4% and excellent stability. This is an innovative approach and the manuscript is well organized. However, there are some flaws in the manuscript. Therefore, a major revision is necessary before it could be published.

1. In the third paragraph, "Residual tensile strain ... stems from the thermal expansion mismatch between perovskites and substrates during the annealing steps ...". The authors use an ultrathin top layer (PDCBT < 8 nm) to balance the tensile/compressive strain. However, the thickness of perovskite layer is estimated to be more than 500 nm (Figure 4a). Why did the author believe this ultrathin top layer (PDCBT) can balance the tensile/compressive strain, which formed between the underneath perovskite layer and the substrate due to mismatched thermal expansion?

2. "The depth-dependent grazing incidence XRD (GIXRD) ... indicate the homogeneous distribution of compressive strain in this perovskite film (Supplementary Fig. 10)". It might be inappropriate by using multi-angle GIXRD in CsPbI₂Br film to probe the homogeneous distribution of compressive strain in this perovskite film. Why do the authors think that the negligible angle change caused by top layer can balance the tensile/compressive strain? How would the angle of GIXRD peak shift without PDCBT treatment?

3. "Room-temperature-prepared CH₃NH₃PbI₃-based solar cells exhibit a PCE of 15.7%". However, in fact, room-temperature-prepared CH₃NH₃PbI₃-based solar cells have reached a PCE of 17.1% which is higher than that of the devices with balanced tensile/compressive strain in this manuscript. (Advanced Materials, 2017, 29(13): 1604695). Therefore, this description in the text may reduce the convinceability of this work.

4. On the calculation of band gap by UV-vis spectrum (Figure S6), please make a recalculation to ensure correct values.

5. As described in the manuscript "(i) ... decrease $\Delta\alpha_{11}$ ". All-inorganic perovskite cannot be room-temperature-prepared due to the phase transformation at low temperature. However, $\Delta\alpha$ can be decreased by using plastic substrates (ref 11). So, would the optimization effect be more obvious if the device was made by plastic substrate?

Reviewer #3 (Remarks to the Author):

The authors developed and employ a strain-compensation strategy: the thermally induced tensile strain in perovskite films is reduced by applying an external compressive strain produced by top functional layer. They found that compressive strain increases the activation energy for ion migration, hence, improving the intrinsic stability of perovskites upon light exposure and temperature. This is a novel approach and interesting for the community. Therefore, I recommend publication after addressing the following points:

- 1) Concerning the homogeneity of the films. Are the films homogeneous in both lateral and vertical direction?
- 2) How thick are the perovskite films? I assume that the perovskite films are fully strained (from both

sides) or is there any relaxation/strain gradient?

3) At the interface of the perovskite and the HTL and ETL layers, I assume there is a natural lattice mismatch. This lattice mismatch induces either tensile or compressive strain by itself which changes/influences the properties of the perovskite film at constant temperature. Can the authors comment on that?

4) I suppose the phase transition in the mixed halide inorganic system is more complicated than the simple CsPbI₃ or CsPbBr₃? Did the authors test their hypothesis for ion migration activation energy on the CsPbI₃ or CsPbBr₃ systems as well? A comparison between the mixed halide system and the non-mixed halide systems would help to drive a universal relation between strain and the ion migration activation energy for total inorganic perovskites.

Response Letter to Reviewers' Comments

For Reviewer #1:

This manuscript describes a strategy to compensate residual tensile strain in perovskite films by introducing an external compressive strain from the hole-transport layer (HTL). The perovskite solar cells (PSCs) achieved a power conversion efficiency (PCE) of 16.4% and retain 96% of their initial PCEs after heating at 85 °C for 1000 hours – the most stable wide-bandgap perovskites (> 1.75 eV) reported so far. Strain engineering is important to further promote the performance of perovskite solar cell. The result is exciting and the strategy to compensate residual tensile strain is novel. Therefore, I suggest accepting the manuscript after some concerns are addressed as follows.

Comment 1: The authors described that they used a substrate with lower coefficient of thermal expansion (α) and the HTL with higher α . So, after annealing, because of the α mismatch between functional layer and substrate, the HTL would bear a tensile strain due to substrate constraint, at the same time the perovskite film would also bears tensile strain. Since both of perovskite film and HTL bear tension from substrate, how can the HTL compensate residual tensile strain in perovskite films? It requires further explanation. Considering the PDCBT hot-casting process. the prepared perovskite film was first preheated to a desired temperature (60 C-120 C) on a hot plate. Is it the heating process that affects the strain in perovskite films?

Response 1:

(i) We clarify this explanation in the response below, and in the manuscript. There is no direct contact between the substrate and the HTL, so the HTL does not bear any tensile or compressive strain from the substrate. The tensile strain only resides in perovskite film due to the substrate constraint. Considering the device architecture substrate/ETL/perovskite/HTL, and each layer's thermal expansion coefficient, the middle perovskite layer would bear a tensile strain from the substrate and a compressive strain from HTL. Therefore, the HTL layer can compensate the residual tensile strain in perovskite films by introducing a compressive strain.

(ii) The perovskite films in this work were prepared at 160°C, which was higher than the preheated temperature (60°C- 120°C) during the HTL casting process. Therefore, we can conclude that the heating process by itself would not affect the strain in perovskite. Jinsong Huang et al. have similarly reported that the strained film was still strained even after being annealed at 100°C for 20 hours due to the strong adhesion of the perovskite to the substrate once the perovskite is formed (Ref. 11: *Sci. Adv.* 2017, 3, eaao5616).

To clarify our statement, we have added relevant discussion:

The residual strain in the perovskite films was insensitive to post-annealing treatment due to the strong adhesion between perovskite and substrate once the perovskite is formed, according to a previous report¹¹. (Page 9)

Comment 2: In the manuscript the correlation between stress (σ) and thermal expansion mismatch is quantified as formula (1):

$$\sigma_{\Delta T} = \frac{E_p}{1 - \nu_p} (\alpha_s - \alpha_p) \Delta T$$

This formula can be used to calculate the stress due to thermal expansion mismatch between monolayer and substrate, is it be appropriate for calculating the stress due to thermal expansion mismatch between bi-layers (perovskite and HTL) and substrate? I think maybe it is better to adopt equations for accurate multilayer estimation.

Response 2: Indeed, this formula has been widely used to quantitatively calculate the stress based on the thermal expansion mismatch of two contacting layers. In this work, we calculate the stress in the perovskite film according to the mismatch for the substrate/perovskite and the perovskite/HTL separately. Then, the net average stress in the perovskite structure when it is situated between the substrate and the HTL (i.e. in an architecture of substrate/perovskite/HTL) can be obtained by the combining the two types of stresses based on the calculated values for substrate/perovskite and for perovskite/HTL. Therefore, both the calculated stress and the net stress are quantitative.

Comment 3: What is the thickness of PDCBT layer? In Figure 4(a), I can evaluate the thickness of PDCBT layer is thinner than TiO₂-Cl layer, about several tens of nanometers. The authors attribute XRD peaks shift to the compressive strain from the HTL. It is doubted that the HTL layer with this thickness can affect the whole perovskite layer that is much thicker and stiffer. I suggest the authors give more evidence to support that the XRD peaks shift is mainly comes from the compressive strain from the HTL. If the XRD peaks shift was mainly comes from the compressive strain from the HTL, can the XRD peaks of perovskite film reverse to tension strain state after the HTL layer is washed out? If possible, the grazing incidence XRD measurement can be used to explore the surface strain change of perovskite film after HTL layer deposition.

Response 3:

(i) To clearly determine the thickness of the PDCBT layer, we have provided the newly-added supplementary Fig. 18. The thickness of PDCBT layer is about 50 nm, consistent with the result obtained from Fig. 4a.

(ii) Although the PDCBT layer is much thinner than the perovskite film, the thermal expansion coefficient of PDCBT is more than an order of magnitude higher than that of the perovskite, providing a large thermal expansion mismatch. Combined with the strong chemical bonding between the PDCBT and the perovskite layers, coating PDCBT on top of the perovskite film at high temperatures can induce a large compressive strain once cooling to room temperature, thus offsetting the tensile strain.

(iii) According to the reviewer's suggestion, we have carried out XRD measurement on the perovskite film after the HTL layer is washed off by chlorobenzene. As shown

in the newly-added supplementary Fig. 11, an obvious shift to higher angle is observed, indicating that the previous tensile strain recovers after the HTL layer is washed away. The above result confirms that the compressive strain stems from the HTL. Moreover, as shown in supplementary Fig. 12, we have applied depth-dependent grazing incidence XRD (GIXRD) to measure the strain distribution in the perovskite film after the HTL layer deposition. It is clear that the compressive strain is distributed throughout the entire perovskite film. The GIXRD results indicate that the residual tensile strain in the perovskite film can be compensated by depositing the HTL atop the perovskite at high temperatures.

In the revised manuscript, we have added the relevant XRD measurements and discussion to make this point clear.

Supplementary Figure 18. SEM top-view image of CsPbI₂Br/PDCBT film.

Supplementary Figure 11. Characterization of strain in perovskite films. (a) XRD patterns of perovskite/PDCBT films fabricated with a PDCBT spin-coating temperature of 120°C, and perovskite films after the PDCBT layer is washed away. (b) Magnified (100) diffraction peaks in the highlighted (blue) region.

Furthermore, the comparison of XRD peaks between the perovskite/PDCBT film and the same perovskite film after the PDCBT is washed away indicates that the compressive strain stems from the PDCBT layer coated at high temperature (Supplementary Fig. 11). (Page 10)

Comment 4: In the DFT simulation, it lacks details with respect to the strain states. Is it under triaxial tension or biaxial? It is better to stimulate the strain state that exhibit in the perovskite film experimentally obtained.

Response 4: The thermally-induced tensile strain in the perovskite films is biaxial, stemming from the substrate. Therefore, we carried out DFT simulations using both

biaxial tensile and compressive strains accordingly.

In the revised manuscript, we have added the strain states in the simulation method section to make this point clear:

Once halide vacancies are formed, we further calculated the relative activation energies for the vacancy-assisted migration of halide ions (E_a) in perovskites under biaxial tensile and compressive strains. (Page 10)

Activation energies for diffusion processes under different biaxial strains were computed from the total energy difference between diffusing species in the ground-state configuration and at the saddle point of the diffusion process⁴⁰. (Page 17)

Comment 5: In supplementary Figure 1e, the strain under 240 °C is about 0.05, such a large strain can fracture the steel, is it reasonable? Please check the number, or it requires a careful explanation.

Response 5: Recently, Sheng Xu et al. reported the strained epitaxial growth of halide perovskite single-crystal thin films on lattice-mismatched halide perovskite substrates (*Nature* 2020, 577, 209). The strain level of FAPbI₃ was as high as 2.4%. McGehee and Dauskardt et al. applied a high external stress ranging from -130 MPa to 130 MPa on perovskite films to investigate the connection between perovskite film stress and the resulting stability (Ref. 12: *Adv. Energy Mater.* 2018, 8, 1802139), though only 50 MPa stress can deform copper. Qihua Xiong et al. reported perovskites that endured about 6%, 4.4% and 5.3% along the c axis, a axis and b axis, respectively (*Sci. Adv.* 2019, 5, eaav9445). Islam et al. found that FAPbI₃ perovskite has a bulk modulus of 16.5 GPa, indicating a much softer lattice of perovskite compared to typical ceramic oxide perovskites (*Chem. Mater.* 2019, 31, 4063). Therefore, according to the above reports, perovskites can endure the large strain due to their soft lattice.

Comment 6: I suggest that the J-V curves under forward and reverse bias of compressive-strain, strain-free and tensile-strain PSCs should be added in supplementary information for the figure 4c.

Response 6: Figure 4c shows the best-performing solar cells of compressive-strain, strain-free, and tensile-strain PSCs. We have provided their corresponding J-V curves under forward and reverse bias in Supplementary Figure 16 to make this point clear.

Comment 7: On Page 14, line 331, what is the specific experiment condition of the thermal stability of PSCs under continuous heating at 85 °C, for example, the humidity of storage environment.

Response 7: The devices measured for thermal long-term stability were kept at 85°C in a nitrogen-filled glovebox.

In the revised manuscript, we have provided the specific experiment condition in the section of Material and device characterization to make this point clear.

Thermal long-term stability tests were carried out by repeating the J-V

characterizations over various times. The devices were kept at 85°C in a nitrogen-filled glovebox. (Page 17)

For reviewer #2:

In this manuscript, the authors demonstrate a strain-compensation strategy to reduce the tensile strain in perovskite films with the aid of top layer (PDCBT). The authors claim this method can balance the tensile/compressive strain transition by tuning the processing temperature and strain of the top layer (PDCBT). The PSCs have a champion PCE of 16.4% and excellent stability. This is an innovative approach and the manuscript is well organized. However, there are some flaws in the manuscript. Therefore, a major revision is necessary before it could be published.

Comment 1: In the third paragraph, “Residual tensile strain ... stems from the thermal expansion mismatch between perovskites and substrates during the annealing steps ...”. The authors use an ultrathin top layer (PDCBT < 8 nm) to balance the tensile/compressive strain. However, the thickness of perovskite layer is estimated to be more than 500 nm (Figure 4a). Why did the author believe this ultrathin top layer (PDCBT) can balance the tensile/compressive strain, which formed between the underneath perovskite layer and the substrate due to mismatched thermal expansion?

Response 1:

(i) The thickness of the PDCBT layer is about 50 nm as seen from the added supplementary Figure 18. The ultrathin PDCBT layer (< 8 nm) is only used to characterize the interaction between PDCBT and the perovskite, due to the shallow probing depth of XPS (~ 10 nm).

(ii) It is known that thermally-induced tensile strain arises in the perovskite layer due to the mismatched thermal expansion coefficients between perovskite and substrate. Our strain-compensation strategy can balance the tensile/compressive strain for two reasons: a) although the top layer HTL is thinner than the underlying perovskite layer, the PDCBT HTL has a much higher thermal expansion coefficient, more than an order of magnitude higher than that of perovskite. This large thermal expansion mismatch offers a large compressive strain when cooling from the high coating temperature of PDCBT to room temperature, which is supported by the XRD patterns of perovskite/PDCBT films fabricated using different PDCBT spin-coating temperatures. b) PDCBT possesses numerous carbonyl anchoring groups and has a very strong mechanical coupling with the perovskite surface, which is supported by XPS measurement. This strong PDCBT/perovskite interface can transfer the compressive strain from PDCBT to the perovskite layer.

(iii) We have carried out XRD measurements on the perovskite film after the HTL layer is washed away by chlorobenzene. As shown in the newly-added supplementary Fig. 11, an obvious shift of the diffraction peak to higher angles is observed,

indicating that the previous tensile strain is recovered after the HTL layer is washed away. The above results confirm that the compressive strain stems from the HTL. Moreover, as shown in supplementary Fig. 12, we have applied depth-dependent grazing incidence XRD (GIXRD) to measure the strain distribution in the perovskite film after the HTL layer deposition. It is clear that the compressive strain is distributed throughout the perovskite film. The GIXRD results indicates that the residual tensile strain in the perovskite film can be compensated by depositing the HTL atop the perovskite at high temperature.

In the revised manuscript, we have added relevant XRD measurements to make this point clear.

Supplementary Figure 11. Characterization of strain in perovskite films. (a) XRD patterns of perovskite/PDCBT film fabricated with a PDCBT spin-coating temperature of 120 °C, and perovskite films after the PDCBT layer is washed away. (b) Magnified (100) diffraction peaks in the highlighted (blue) region.

Furthermore, the comparison of XRD peaks between perovskite/PDCBT film and the same perovskite film after PDCBT is washed away indicates that the compressive strain stems from the PDCBT layer coated at high temperature (Supplementary Fig. 11). (Page 10)

Comment 2: “The depth-dependent grazing incidence XRD (GIXRD) ... indicate the homogeneous distribution of compressive strain in this perovskite film (Supplementary Fig. 10)”. It might be inappropriate by using multi-angle GIXRD in CsPbI₂Br film to probe the homogeneous distribution of compressive strain in this perovskite film. Why do the authors think that the negligible angle change caused by top layer can balance the tensile/compressive strain? How would the angle of GIXRD peak shift without PDCBT treatment?

Response 2: Grazing incidence XRD (GIXRD) has been demonstrated to be a powerful technique for characterizing the structural variation of a sample, where different depths can be probed by changing the incident angle of the X-rays (Supplementary Figure 9a). By using incident angles varying from 0.1° to 1°, we are able to reveal the structural differences in the perovskite layer at a theoretical probing depth of ~30 nm to 3000 nm (*ACS Appl. Mater. Interfaces* 2017, 9, 23141; *Adv. Funct. Mater.* 2015, 25, 2892). Considering the thickness of the as-deposited perovskite layer

(~550 nm), the GIXRD patterns collected at incident angles of 0.1° , 0.3° , 0.5° , 0.8° and 1° exhibit structural information from the surface to the bottom of perovskite layer.

(i) As shown in Supplementary Figure 12 for the characterization of the perovskite/PDCBT film, the small variation of the lattice constant at different incident angles indicates the homogeneous distribution of compressive strain throughout the perovskite film.

(ii) As for the GIXRD peak shift without the PDCBT treatment, we have provided GIXRD patterns of perovskite films as a function of incident angle (Supplementary Figure 9). The results demonstrate the homogeneous distribution of tensile strain in the perovskite film.

In the revised manuscript, we have modified the supplementary Figure 9a and relevant discussion to make this point clear.

Supplementary Figure 9. (a) Scheme of the grazing incidence XRD (GIXRD) measurement.

By using incident angles varying from 0.1° to 1° , we are able to reveal the structural differences in the perovskite layer at a theoretical probing depth of ~30 nm to 3000 nm^{1,2}.

Comment 3: “Room-temperature-prepared $\text{CH}_3\text{NH}_3\text{PbI}_3$ -based solar cells exhibit a PCE of 15.7%”. However, in fact, room-temperature-prepared $\text{CH}_3\text{NH}_3\text{PbI}_3$ -based solar cells have reached a PCE of 17.1% which is higher than that of the devices with balanced tensile/compressive strain in this manuscript. (Advanced Materials, 2017, 29(13): 1604695). Therefore, this description in the text may reduce the convinceability of this work.

Response 3: According to the reviewer’s suggestion, we have read the work published in *Adv. Mater.* 2017, 1604695. There is no doubt that the room-temperature-prepared $\text{CH}_3\text{NH}_3\text{PbI}_3$ -based solar cells have a higher PCE of 17.1% than that of our CsPbI_2Br -based solar cells (16.4%) with balanced tensile/compressive strain. However, it may be inappropriate to compare the PCEs of $\text{CH}_3\text{NH}_3\text{PbI}_3$ and CsPbI_2Br solar cells directly due to their different bandgaps and thermal stabilities.

The theoretical Shockley-Queisser efficiency limits for $\text{CH}_3\text{NH}_3\text{PbI}_3$ (1.64 eV) and CsPbI_2Br (1.92 eV) solar cells are roughly 30% and 23%, respectively. Specifically, our compressively-strained CsPbI_2Br -based devices have reached roughly 72% of the theoretical efficiency limit while room-temperature-processed $\text{CH}_3\text{NH}_3\text{PbI}_3$ devices have reached only 57%. Moreover, the current record for the hybrid organic-inorganic perovskite solar cells has been reached to 25.2%, much higher than that of its room-temperature-prepared devices (17.1%). The large gap between state-of-art $\text{CH}_3\text{NH}_3\text{PbI}_3$ devices and room-temperature-processed devices indicates the limitation of room-temperature processing. In contrast, the PCE of 16.4% for our compressively-strained CsPbI_2Br -based devices is nearly the highest efficiency for wide-bandgap perovskite (> 1.75 eV) cells and also the most stable wide-bandgap PSC reported so far. Therefore, the excellent performance of compressively-strained devices fully demonstrates the credibility of our work. In the revised manuscript, we have cited the mentioned important literature as reference 14 and added relevant discussion to make this point clear.

Considering the wide bandgap of CsPbI_2Br (1.92 eV), the performance of our compressive-strain devices is comparatively superior to that of strain-free $\text{CH}_3\text{NH}_3\text{PbI}_3$ -based devices with a bandgap of 1.64 eV fabricated at room temperature¹⁴.(Page 14)

Comment 4: On the calculation of band gap by UV-vis spectrum (Figure S6), please make a recalculation to ensure correct values.

Response 4: We have carefully rechecked our calculations and found that there was a mistake with the values of $(\alpha h\nu)^2$. The correct order of magnitude is 10^8 rather than 10^{18} . However, this mistake does not affect the determination of bandgap. The calculated bandgap of 1.91 eV is consistent with previously reported results for CsPbI_2Br . In the revised manuscript, we have modified the Supplementary Figure 6a to make this point clear.

Supplementary Figure 6. Optical characterizations of CsPbI_2Br film formed at 160 °C. (a) Absorption spectrum of CsPbI_2Br film. Inset: Tauc plot for CsPbI_2Br film to determine the bandgap of CsPbI_2Br .

Comment 5: As described in the manuscript “(i) ... decrease $\Delta\alpha_{11}$ ”. All-inorganic

perovskite cannot be room-temperature-prepared due to the phase transformation at low temperature. However, $\Delta\alpha$ can be decreased by using plastic substrates (ref 11). So, would the optimization effect be more obvious if the device was made by plastic substrate?

Response 5: According to formula 1, due to the negligible thermal expansion mismatch between the perovskite and the plastic substrate, coating the perovskite onto a plastic substrate is indeed an effective method to reduce the tensile strain in perovskite films. Recently, McGehee and Dauskardt et al. successfully reduced perovskite tensile strain through depositing films on polycarbonate (PC) substrates with a high thermal expansion coefficient of $6.5 \times 10^{-5} \text{ K}^{-1}$ (Ref. 12: *Adv. Energy Mater.* 2018, 8, 1802139). Similarly, Jinsong Huang et al. used polystyrene (PS) to regulate the tensile strain in perovskite films (Ref. 11: *Sci. Adv.* 2017, 3, eaao5616). However, plastic substrates limit the processing temperature of perovskites, and all-inorganic perovskites have to be processed under high temperatures to stabilize the black cubic perovskite phase. Thus, the use of plastic substrates compromises the device performance. The record efficiency of flexible perovskite solar cells (19.38%; *Adv. Funct. Mater.* 2019, 29, 1902974) is still far lower than that of the rigid devices (25.2%).

For reviewer #3:

The authors developed and employ a strain-compensation strategy: the thermally induced tensile strain in perovskite films is reduced by applying an external compressive strain produced by top functional layer. They found that compressive strain increases the activation energy for ion migration, hence, improving the intrinsic stability of perovskites upon light exposure and temperature. This is a novel approach and interesting for the community. Therefore, I recommend publication after addressing the following points:

Comment 1: Concerning the homogeneity of the films. Are the films homogeneous in both lateral and vertical direction?

Response 1: We have performed top-view SEM-EDS elemental mapping measurements to characterize the film homogeneity in the lateral direction. As shown in the newly added Supplementary Figure 7, the elemental mappings of Cs, Pb, I, and Br show that there is no elemental aggregation, indicating the homogeneity of the perovskite film in the lateral direction. As for the vertical direction, we have provided the cross-section SEM-EDS line scan profiles and elemental maps in Supplementary Figure 8. The results indicate the homogeneous distribution of compositional elements in the vertical direction. In the revised manuscript, we have modified the Supplementary Figure 7 to make this point clear.

Supplementary Figure 7. Material characterization of CsPbI₂Br film formed at 160 °C. (a) SEM top-view image of CsPbI₂Br film. (b) AFM image of CsPbI₂Br film. EDS elemental maps of (c) Cs, (d) Pb, (e) I, and (f) Br.

Comment 2: How thick are the perovskite films? I assume that the perovskite films are fully strained (from both sides) or is there any relaxation/strain gradient?

Response 2:

(i) The thickness of the perovskite film is about 550 nm, as shown in Figure 4a.

(ii) Grazing incidence XRD (GIXRD) has been demonstrated to be a powerful technique for characterizing the structural variation of a sample, where different depths can be probed by changing the incident angle of the X-rays (Supplementary Figure 9a). By using incident angles varying from 0.1° to 1°, we are able to reveal the structural differences in the perovskite layer at a theoretical probing depth of ~30 nm to 3000 nm (*ACS Appl. Mater. Interfaces* 2017, 9, 23141; *Adv. Funct. Mater.* 2015, 25, 2892). Considering the thickness of the as-deposited perovskite layer (~550 nm), the GIXRD patterns collected at incident angles of 0.1°, 0.3°, 0.5°, 0.8° and 1° exhibit structural information from the surface to the bottom of perovskite layer.

(iii) To characterize the strain distribution in the perovskite film, we have performed the GIXRD measurements as shown in Supplementary Figure 9 and 12. The neglectable variation of the lattice constant from the top surface to the bottom of the film indicates that the perovskites are fully strained from both sides, and there is no relaxation or strain gradient.

Comment 3: At the interface of the perovskite and the HTL and ETL layers, I assume there is a natural lattice mismatch. This lattice mismatch induces either tensile or compressive strain by itself which changes/influences the properties of the perovskite film at constant temperature. Can the authors comment on that?

Response 3: We agree with the reviewer on the lattice mismatch between perovskite and HTL/ETL layers. One of the key points for the strain formation in perovskite is the strong interaction between the perovskite and the HTL/ETL layers. These strong interactions, and the lattice mismatch at room temperature, lead to the tensile or

compressive strain in the perovskite. We have provided a detailed discussion about the strain formation on page 6 (Fig. 1c). As the reviewer has said, the tensile or compressive strain would directly affect the properties of the perovskite film. In this work, we have investigated the correlation between the strain and the properties of the perovskite from the following aspects:

(i) Compared with intrinsic (strain-free) perovskites, tensile strain decreases the formation energy of halide vacancies, while compressive strain increases their formation energy, as shown in Supplementary Figure 13;

(ii) Tensile strain accelerates the ion migration, while compressive strain decelerates ion migration, improving the intrinsic stability of the perovskite, including photostability and thermal stability, as shown in Figure 3 and Supplementary Figure 13. Additionally, Qi Chen et al. reported the strain impact on the carrier dynamics of perovskites, which stems from the strain induced energy band bending of the perovskite (*Nat. Commun.* 2019, 10, 815). We have cited this paper as ref. 13.

Comment 4: I suppose the phase transition in the mixed halide inorganic system is more complicated than the simple CsPbI₃ or CsPbBr₃? Did the authors test their hypothesis for ion migration activation energy on the CsPbI₃ or CsPbBr₃ systems as well? A comparison between the mixed halide system and the non-mixed halide systems would help to drive a universal relation between strain and the ion migration activation energy for total inorganic perovskites.

Response 4: As the reviewer said, the phase transition in mixed halide inorganic perovskites may be more complicated than the simple CsPbI₃ or CsPbBr₃ compositions. The conclusion about the ion migration activation energies obtained from CsPbI₂Br under different strains might be also suitable for CsPbI₃ or CsPbBr₃. To test this hypothesis, we calculated the activation energies for halide-vacancy-assisted migration in CsPbI₃ and CsPbBr₃ under tensile and compressive strains, as shown in Table R1. The activation energies for halide ion migration in strain-free CsPbI₃ and CsPbBr₃ are 0.621 eV and 0.713 eV, respectively. Lower activation energies of 0.504 eV and 0.612 eV are found in CsPbI₃ and CsPbBr₃ under a tensile strain of 1.5%, while higher energies of 0.693 eV and 0.853 eV are found under a compressive strain of -1.5%. Therefore, the above results indicate that compressive strain enhances the ion migration activation energy for fully inorganic perovskites, whereas tensile strain decreases the ion migration activation energy.

Table R1 Calculated activation energies for halide-vacancy-assisted migration in CsPbBr₃, CsPbI₂Br, and CsPbI₃ under biaxial tensile strain, no strain and compressive strain.

Materials	E_A (eV)		
	1.5% (tensile strain)	0% (no strain)	-1.5% (compressive strain)
CsPbBr ₃	0.612	0.713	0.853
CsPbI ₂ Br	0.547	0.667	0.794

$CsPbI_3$	0.504	0.621	0.693
-----------	-------	-------	-------

REVIEWERS' COMMENTS:

Reviewer #1 (Remarks to the Author):

I am almost satisfied with the efforts and modifications made by the reviewer. With respect to Reviewer#1 Comment#5 and Reviewer#2 Comment#1, I suggest the authors elaborate the explanations in the main text rather than the rebuttal letter only.

Reviewer #2 (Remarks to the Author):

The authors have seriously answered the questions raised by the reviewers. I recommend the revised manuscript to be published as is.

Reviewer #3 (Remarks to the Author):

The authors addressed all my concerns. I am pleased about their answers and can now endorse the publication of their manuscript.

Response Letter to Reviewers' Comments

Reviewer #1 (Remarks to the Author) :

I am almost satisfied with the efforts and modifications made by the reviewer. With respect to Reviewer#1 Comment#5 and Reviewer#2 Comment#1, I suggest the authors elaborate the explanations in the main text rather than the rebuttal letter only.

We thank the referee for a constructive review process. According to the referee's suggestion, we have added the relevant explanations in the main text.

For reviewer#1 comment#5

This high thermal stability enables all-inorganic perovskites to withstand higher annealing temperatures (Fig. 1b), while also inducing a large amount of stress considering the soft lattice of perovskites that can endure large strain (Fig. 1d and Supplementary Fig. 1), thereby facilitating the investigation of strain control. (Main text, page 5)

For reviewer#2 comment#1

We have provided detailed explanations of our strain-compensation strategy. (Main text, page 6). The following newly added XRD measurements further verify the strategy from experiment.

Furthermore, the comparison of XRD peaks between perovskite/PDCBT film and the same perovskite film after PDCBT is washed away indicates that the compressive strain stems from the PDCBT layer coated at high temperature (Supplementary Fig. 11). (Main text, page 8)

Reviewer #2 (Remarks to the Author):

The authors have seriously answered the questions raised by the reviewers. I recommend the revised manuscript to be published as is.

We thank the referee for a constructive review process.

Reviewer #3 (Remarks to the Author):

The authors addressed all my concerns. I am pleased about their answers and can now endorse the publication of their manuscript.

We thank the referee for a constructive review process.